# Silicon Nanofluidic Membrane for Electrostatic Control of Drugs and Analytes Elution

**DOI:** 10.3390/pharmaceutics12070679

**Published:** 2020-07-19

**Authors:** Nicola Di Trani, Antonia Silvestri, Yu Wang, Danilo Demarchi, Xuewu Liu, Alessandro Grattoni

**Affiliations:** 1Department of Nanomedicine, Houston Methodist Research Institute, Houston, TX 77030, USA; nditrani@houstonmethodist.org (N.D.T.); antonia.silvestri@polito.it (A.S.); ywang2@houstonmethodist.org (Y.W.); xliu@houstonmethodist.org (X.L.); 2University of Chinese Academy of Science (UCAS), Shijingshan, 19 Yuquan Road, Beijing 100049, China; 3Department of Electronics and Telecommunications, Polytechnic of Turin, 10129 Turin, Italy; danilo.demarchi@polito.it; 4Department of Surgery, Houston Methodist Hospital, Houston, TX 77030, USA; 5Department of Radiation Oncology, Houston Methodist Hospital, Houston, TX 77030, USA

**Keywords:** electrostatic gating, nanofluidic diffusion, controlled drug release, silicon membrane, smart drug delivery

## Abstract

Individualized long-term management of chronic pathologies remains an elusive goal despite recent progress in drug formulation and implantable devices. The lack of advanced systems for therapeutic administration that can be controlled and tailored based on patient needs precludes optimal management of pathologies, such as diabetes, hypertension, rheumatoid arthritis. Several triggered systems for drug delivery have been demonstrated. However, they mostly rely on continuous external stimuli, which hinder their application for long-term treatments. In this work, we investigated a silicon nanofluidic technology that incorporates a gate electrode and examined its ability to achieve reproducible control of drug release. Silicon carbide (SiC) was used to coat the membrane surface, including nanochannels, ensuring biocompatibility and chemical inertness for long-term stability for in vivo deployment. With the application of a small voltage (≤ 3 V DC) to the buried polysilicon electrode, we showed in vitro repeatable modulation of membrane permeability of two model analytes—methotrexate and quantum dots. Methotrexate is a first-line therapeutic approach for rheumatoid arthritis; quantum dots represent multi-functional nanoparticles with broad applicability from bio-labeling to targeted drug delivery. Importantly, SiC coating demonstrated optimal properties as a gate dielectric, which rendered our membrane relevant for multiple applications beyond drug delivery, such as lab on a chip and micro total analysis systems (µTAS).

## 1. Introduction

Chronic pathologies affect nearly half of the population worldwide [1,2] and represent one of the leading causes of death and disability [3]. Management of chronic conditions is challenged by co-morbidities [4], poor adherence to treatment [5], and a lack of therapeutic technologies suitable to address the complexity of the disease [6]. Long-acting controlled therapeutic administration represents a promising strategy for medical conditions requiring repeated daily dosing [7,8]. In view of this, long-acting platforms for sustained drug release have been developed, leading to significant improvements in the management of conditions, such as hormone deficiency and infectious diseases [9,10,11]. However, the pathophysiology of most chronic diseases is determined by circadian biological cycles [12], which have a significant impact on the efficacy of treatment and associated adverse effects [13]. This is the case for pathologies, such as diabetes and metabolic disorders [14], hypertension, psychiatric and neurodegenerative conditions [15], rheumatoid arthritis [16], and chronic pain [17], to name a few, where the timing of drug administration is key to elicit the intended therapeutic effect.

Advanced technologies enabling personalized adjustments of therapeutic administration, both in time and dose, represent a desirable but unmet clinical need [18,19]. Ideally, these technologies should incorporate a drug delivery mechanism that can be rapidly and easily tuned to release the required dosage, at the right time, without requiring continuous external stimuli. Further, they should allow for pre-programmed dosing schedules as well as remote control capabilities, to enable healthcare providers to adjust medication through telemedicine approaches [20]. Devices with such capabilities could eradicate treatment compliance issues and dramatically improve the therapeutic index and the quality of life of patients, while substantially reducing healthcare expenditure.

Current approaches developed for controlled drug administration are based on modulation of permeability of membranes via sustained external stimuli. These systems mostly rely on polymeric membrane architectures and achieve changes in pore size and conformation via temperature variation triggered by a magnetic field [21], near-infrared irradiation [22,23,24], or ultrasound [25]. Other devices use a magnetic field to reversibly or irreversibly obstruct the pores of a membrane using microparticles [26] or low melting temperature polymers [27]. Albeit promising, these strategies are limited by the need for continuous external activation and associated cumbersome external equipment. Electrical actuation offers a solution to these limitations, enabling control via miniaturized circuitry and low energy radio-frequency (RF) communications. In this context, various technologies have been created, either integrating gate electrodes [28] or polypyrrole (PPy) [29,30] on anodic aluminum oxide (AAO) membranes. However, polydispersity in pore size common to AAO membranes represents a limitation to achieve fine control of drug release [28,31].

In this study, we investigated the performance of a silicon nanofluidic membrane that uses electrostatic gating [32] to modulate the transport of charged molecules by modifying nanochannel permeability. Microfabricated using standard semiconductor manufacturing techniques, this membrane features hundreds of thousands of identical slit-nanochannels geometrically distributed across the membrane surface to maximize porosity while maintaining mechanical integrity. A buried polysilicon layer extends over the entire nanochannel surface and acts as a single distributed gate electrode. An outermost layer of biocompatible silicon carbide (SiC) is adopted to bury and insulate the gate electrode and minimize leakage while providing chemical inertness for applications in vivo or in contact with biological fluids. SiC insulation properties were studied in comparison with silicon dioxide (SiO_2_), the most common gate dielectric material in metal–oxide–semiconductor field-effect transistor (MOSFET). Further, energy consumption leakage current and gating performance were assessed at different gate potentials. Finally, we adopted two relevant model analytes—methotrexate and quantum dots—to assess the in vitro transport modulation performances. Methotrexate represents an important therapeutic agent commonly used for rheumatoid arthritis [33], whereas quantum dots are adopted for a variety of biomedical imaging applications as well as drug delivery and theranostics [34,35]. In light of the promising results and the ease of integration within implantable devices, our gated membrane might constitute a promising step forward in the development of flexible technologies for the treatment of chronic diseases. Further, our nanofluidic technology could be adopted in other applications for lab on a chip [36] and micro total analysis systems (µTAS) devices for electrokinetic separation processes [37], bio-sample sorting and analysis [38], among others.

## 2. Materials and Methods

### 2.1. Nanofluidic Membrane Fabrication

Silicon membranes fabrication was performed using standard semiconductor techniques. The fabrication process is described step-by-step elsewhere [39]. Briefly, a dense array of nanochannels (500 nm width, 6 µm length) was obtained by vertically etching via deep reactive ion etching (DRIE) the device layer (10 µm) of a silicon on insulator (SOI) wafer (total thickness 411 µm). The etching was stopped at the middle oxide layer (1 µm). The handle wafer (400 µm) on the opposite side of the SOI was etched using DRIE up to the oxide layer to create a hexagonal pattern of densely packed circular microchannels. To connect the nanochannels to the microchannels, the buried oxide layer was etched by a buffered oxide etchant solution (BOE). The resulting nanochannels size (770 nm) was reduced by three subsequent processes. First, wet thermal oxidation generated a 175 nm layer of SiO_2_, and then a 121 nm layer of polycrystalline silicon (poly-Si) was obtained via low-pressure chemical vapor deposition (LPCVD). Plasma-enhanced chemical vapor deposition (PECVD) was then used for the silicon carbide coating (SiC, 64 nm). Electrode pads were exposed via selective etching of SiC via fluorine-based RIE. Wafers were then diced into individual membranes (ADT 7100 Dicing Saw, Advanced Dicing Technologies, Zhengzhou, China), obtaining individual silicon membranes of 6 mm × 6 mm × 411 µm. Each membrane featured a total of 278,600 identical slit nanochannels 10 µm long and 6 µm wide. Nanochannels were arranged in groups of 1400, where each group led to one circular microchannel on the opposite side of the chip. Finally, microchannels were geometrically organized in a hexagonal pattern to maximize porosity and structural integrity. In this study, membranes with a final layer of SiO_2_ were also used and obtained by wet thermal oxidation of poly-Si.

### 2.2. Assessment of Membrane Structure

Morphological assessment and characterization of the nanochannel multi-layer structure were performed via scanning electron microscopy (SEM). A focused ion beam (FIB) system FEI 235 (Nanofabrication facility of the University of Houston, Houston, TX, USA) was used to simultaneously create nanochannels’ cross-sections and acquire images. Gallium ion milling was performed on the micromachine parts of the membrane and to expose the nanochannel cross-section. Imaging was then performed at a 52° angle. 

### 2.3. Electrode Connection

Electrical wires (36 AWG, McMaster Carr, Elmhurst, IL, USA) were epoxied to the membrane pads using a silver-based conductive adhesive (H20E, Epoxy Technology, Inc., Billerica, MA, USA) and cured at 150 °C for 1 h. Electrode insulation was achieved by applying a thin layer of UV epoxy (OG116, Epoxy Technologies, Inc. Billerica, MA, USA) over the conductive pad and UV-curing (UVP UVL-18 EL Series, Analytik Jena US LLC, Upland, CA, USA) for 120 min. The correct electrode connection was tested by measuring the resistance between the two connection pads with a Fluke 177 True RMS Multimeter (Fluke Corporation, Everett, WA, USA).

### 2.4. Electrochemical Characterization

A custom dual-reservoir polymethyl methacrylate (PMMA) apparatus [39] was employed to perform electrochemical measurements. Membranes were clamped between the two 2 mL reservoirs of the testing apparatus, each containing two Ag/AgCl electrodes (64-1313, Harvard Apparatus, Holliston, MA, USA). All measurements were performed in PBS, except for conductance studies, where KCl solutions at different concentrations (from 10^−6^ to 10^−1^ M) were used. A benchtop electrochemical tester (CH Instruments, Inc. 660E, Austin, TX, USA) was used in either 3 or 4 electrode configurations.

Impedance was measured with a 4-electrodes configuration. A 50 mV perturbation signal was applied through the electrochemical analyzer within a frequency window from 10 mH to 10 kH. The measurements were performed with a superimposed DC voltage in the range −3 V to 3 V in steps of 1 V. Fittings to a Randles cell model were performed with the CHI 660E software (CH Instruments, Inc. 660E, Austin, TX, USA). 

Leakage current was measured with a 3-electrodes configuration. Voltages were applied using the CHI 660E between the electrode pad and Ag/AgCl electrodes in solution at a distance of ~1 cm from the membrane. Measurements were performed in the −3 V to +3 V range in 1 V steps, and each step lasted for 120 s, allowing for transient phenomena to resolve and obtain a stable measurement.

For conductance experiments, we employed a 4-electrodes configuration. Measurements were performed for KCl concentrations from 1 μM to 100 mM from the lower to the highest ionic strength. Reservoirs were rinsed with deionized water for 1 min, and the solution was replaced after each measurement. Steps of 400 mV were applied using the CHI 660E from −2 V to 2 V, with 30 s pauses to exhaust possible transient effects. Conductance measurements were performed with a floating gate, and the values calculated for each step and averaged.

Cyclic voltammetry measurements were conducted with a 3-electrodes configuration and a scanning rate of 50 mV/s within the interval −2 V to 2 V. Electrochemical measurements were carried out on membranes with a final dielectric layer of both SiC and SiO_2_.

### 2.5. In Vitro Release Modulation

In vitro release modulation experiments were performed employing a custom dual-reservoir device described in detail elsewhere [40]. Nanochannel membranes were individually clamped between a 250 µL drug reservoir and a UV-Vis transparent macro-cuvette serving as the sink reservoir. Two O-rings were used to prevent fluid leakage between membranes and the reservoir. Fluid evaporation was prevented by sealing a drug reservoir with biocompatible silicone plugs (McMaster Carr, Elmhurst, IL, USA).

Experiments were performed using SiC-coated membranes with ~300 nm nanochannels. To ensure proper channel wetting, membranes were immersed in isopropyl alcohol for 1 h and then rinsed three times in deionized H_2_O. Membranes were then placed overnight in 0.01 × PBS or 1 × PBS in preparation for quantum dots and methotrexate release, respectively. Sink reservoirs (4.45 mL) were filled with matching PBS solutions. After fixture assembly, the source reservoir was loaded with either 1 mg/mL 0.01 × PBS solution of quantum dots (CdTe core-type, COOH functionalized, 777978-10MG, Sigma Aldrich, St. Louis, MO, USA) or 2.5 mg/mL PBS solution of methotrexate (13960, Cayman Chemical, Ann Arbor, MI, USA). Both molecules possess a negative charge at pH 7.4, with methotrexate presenting a stable −2q charge (−3.2 × 10^−19^ C) and quantum dots having a charge that ranges from −5q to −15q depending on pH and ionic strength [41]. Methotrexate has a molar mass of 454 Da and an estimated diameter of 1.6 nm [42], while quantum dots have an estimated molar mass of 200 kDa and an estimated diameter of 4.7 nm [43]. An Ag/AgCl reference electrode (Harvard Apparatus, Holliston, MA, USA) was used and placed in the source drug reservoir.

Absorbance measurements of every sample were performed at 5 min intervals using a custom UV-vis spectrophotometer apparatus consisting of a robotic carousel [44] connected to an Agilent Cary 50 spectrophotometer (Agilent, Technologies, Santa Clara, CA, USA). Sink solution homogeneity was maintained by constant magnetic stirring (600 rpm). Methotrexate absorbance was measured at 373 nm, while quantum dots at 240 nm. An electrical potential (0, −1.5, or −3 V DC) was applied between the Ag/AgCl and the membrane electrodes through a waveform generator (33522A, Keysight Technologies, Santa Clara, CA, USA). Passive (0 V) and active (−1.5 or −3 V) phases were alternated at regular intervals. For methotrexate, phases were alternated every 6 h between passive and active (0 and −3 V DC, respectively). For quantum dots, 12 h passive phases were alternated with 8 h of active applied potential (−1.5 V).

### 2.6. Statistical Analysis

Statistical analysis was performed using GraphPad Prism 8 (version 8.1.1; GraphPad Software, Inc., San Diego, CA, USA). Mean ± SD values were calculated for all results. Further statistical significance was assessed, adopting the two-tailed paired t-tests (** *p* ≤ 0.01; **** *p* ≤ 0.0001). Cumulative releases were split into phases, and each fitted by a first-order polynomial (MATLAB^®^ polyfit, MathWorks, Natick, MA, USA). Slopes of cumulative release curves were normalized and displayed as a percentage of the passive release profiles.

## 3. Results and Discussion

### 3.1. Nanofluidic Membrane

Prior to investigating the electrical performance of the membranes, we sought to analyze the quality of the membrane fabrication process (Figure 1). Individual silicon membranes were first visually inspected to assess integrity. Figure 1A shows a stereomicroscope picture of a single membrane, highlighting the conductive electrode pads at the top right and bottom left edges. The hexagonal arrangement of microchannels allowed us to maximize packing density without compromising mechanical robustness. By measuring transmembrane nitrogen gas flow and adopting our predictive model for nanofluidic gas transport [45], we obtained an indirect measurement for the size of nanochannels (~300 nm). Sample membranes were further analyzed with SEM imaging. Figure 1B shows the tightly packed nanochannel arranged in arrays of 19 rows and 96 columns with a horizontal pitch of 2 µm and a vertical pitch of 10 µm. No macroscopic defects or pinholes were observed across wafers, which indicated that the fabrication protocol was repeatable.

The analysis of the membrane cross-sections obtained via FIB milling was performed to evaluate the uniformity of layer deposition at different nanofabrication steps. SiO_2_ growth via thermal oxidation resulted in a highly uniform layer along the whole length of the vertical nanochannels (Figure 1C). Thermal oxidation is a slow process that enables precise control over layer thickness. Thus, it allowed us to accurately and homogeneously reduce the size of nanochannels. The subsequent deposition of poly-Si (Figure 1D) was used to create a gate electrode that coats the whole nanofluidic structure with the objective of maximizing the electrostatic gating performances. Uniform poly-Si deposition through the chemical vapor deposition-based (CVD) process in high-aspect-ratio hollow structures can be challenging. However, our imaging analysis showed that the deposited layer was uniform (Figure 1D), except for a slight increase in thickness at the nanochannel outlet (bottom right). Finally, a thin layer of SiC (Figure 1E) was used to coat the conductive poly-Si and act as an insulating and chemical inert layer. Despite the high-aspect-ratio of the slit nanochannels, the deposition of SiC was also achieved with good uniformity (Figure 1E). Slight material accumulations at the inlet and outlet of nanochannels were expected. While we did not generate these intentionally, we noted that a local restriction at the nanochannel extremities could improve gating performance.

All materials used for the fabrication of the nanofluidic membrane have previously been demonstrated to be biocompatible using ISO 10993 standards by Kotzar et al. [46]. A subset of these materials has also been investigated in vivo in rodents and has shown biocompatibility and low biofouling [47]. Moreover, in our fabrication protocol, silicon carbide completely encapsulates the membrane and, therefore, is the only material exposed to the environment. Silicon carbide was specifically chosen for this encapsulation purpose as it’s considered a versatile material for biomedical applications where extended exposure to physiological fluids is needed [48,49]. Additionally, in vivo biocompatibility of SiC was demonstrated by Cogan et al. [50], who subcutaneously implanted SiC discs in New Zealand White rabbit, the histological evaluation showed no chronic inflammatory response, and a capsule thickness comparable to controls was found.

Furthermore, silicon carbide has previously been shown to offer reduced biofouling when compared to other biocompatible materials, such as silicon or silicon dioxide [51]. Although complete protection against protein adsorption could not be achieved [52], we previously showed that biofouling did not negatively affect the function of our devices. Specifically, nanofluidic membranes, similar to the one presented in this study, have been used in-vivo in rats for up to 6 months [53] and in non-human primates for up to 4 months [54] with no alteration of drug release from biofouling or fibrotic tissue encapsulation.

When compared to membrane architectures previously developed in our lab [55,56,57], this membrane presented a less cumbersome fabrication process, thanks to the direct alignment of nanochannels and microchannels [58,59]. Further, a substantially higher nanochannels density [55,60] was achieved. In contrast with other gated membrane based on porous alumina (AAO) [29], presenting an irregular pore size distribution [28], our structure achieved a monodispersed nanochannel size that could aid in better control of molecular transport. In its current configuration, featuring 278,600 nanochannels, our membrane configuration was designed to achieve high mass transport rates per unit surface area. This is typically preferable in the context of implantable drug delivery application, where miniaturization is a requirement [39]. However, in light of its modular structure, the same fabrication process could be employed to create alternative configurations with a different number of channels for adoption in electrokinetic-enabled molecular manipulation or sorting applications. For these purposes, the large gate electrode surface area might provide increased electrostatic control of fluid molecules as compared to common Polydimethylsiloxane-glass (PDMS-glass) systems [61,62] which feature localized gate electrodes.

### 3.2. Solid–Liquid Interface, SiO_2_ vs SiC

To evaluate SiC properties as a gate dielectric in contact with ionic solutions, we compared its insulation performance to SiO_2_, which is a broadly used gate dielectric in solid electronics [63]. SiO_2_ and other metal oxides, such as alumina and hafnium dioxide, owe their success to their high dielectric constants that allow for low leakage currents. Even though these materials excel in solid electronic manufacturing, they either lack biocompatibility or chemical inertness and durability in aqueous environments [50].

Leakage current measurements (Figure 2A) performed with our membranes did not show substantial differences between SiO_2_ and SiC, except for 3 V. However, the steep increase in leakage observed for SiC between 2 and 3 V, emphasized by the electrolytic solution environment, suggests that the molecular arrangement in the dielectric layer is not ideal [64]. The literature on gate dielectric leakage in ionic solutions is scarce, and the available models for a solid-state field-effect transistor (FET) are unable to account for the effect of the electrolyte solution environment. In aqueous solutions, currents in the order of µA were measured for electric fields as low as 0.5 MV cm^−1^ (Figure 2A). In contrast, for solid-state FET, currents in the order of magnitude of µA are only expected for electric fields greater than 15 and 2 MV cm^−1^ for SiO_2_ and SiC, respectively. High leakage currents are usually attributed to the formation of conductive filaments within the oxide, whereby electrons are trapped and form clusters within defects in the material. When clusters are at tunneling distance, a conductive path can form, leading to high leakage currents [65,66]. The proportional increase in leakage currents at increasing ionic strength of the solution, previously reported by this group [39], provides further support for this phenomenon.

In the voltage range between −2 and 2 V, SiC and SiO_2_ exhibited similar values of leakage currents. Thus, to closer investigate differences in performances, we used cyclic voltammetry (CV). As compared to SiO_2_-coated membranes, lower currents were measured for SiC at each applied voltage (Figure 2B). Interestingly, we observed a non-linear proportional relationship between voltage and current for both materials. SiC exhibited a steep increase in current for voltages higher than 1 V in absolute value. This suggested that for small applied voltages, no faradaic currents occurred, and the material behaved almost as an ideal capacitor. For voltages above ± 1 V, electrochemical reactions between the surface groups (C, SiO^−^) and reactive species in the electrolyte solution (Cl^−^, HO^−^) led to increased currents.

In contrast, the significant current increase observed for the leakage currents (Figure 2A) for voltages over 2 V was likely related to material deterioration and conductive filament formation. The asymmetry between results obtained with positive and negative voltages provided further support for this theory. Higher currents for negative applied voltages were observed in both measurements. For negative voltages, positive species were attracted to the surface. The percolation model suggests that in the presence of strong electrostatic attraction, protons can diffuse in the insulator, starting a percolating path that can lead to the formation of a conductive filament [65]. Instead, for positive potentials, proton repulsion may cause a reversible interruption of the conductive filament, effectively decreasing leakage [67]. Additionally, the difference in hysteresis between the two CV profiles (Figure 2B) was suggestive of differences in surface charge accumulation between the two materials. A thinner CV profile usually correlates with low charge accumulation. Collectively, the results showed that SiC suffered lower leakage currents in the −2 V to 2 V range, exhibiting better insulation performance than SiO_2_.

### 3.3. Electrochemical Characterization: Conductance

To further investigate the surface properties of SiC, we performed conductance measurements of SiC-coated membranes in the ionic concentration range between 1 µM and 100 mM. We employed a custom fixture [39] that allowed us to limit wetting to the nanochannel part of the membrane. The results are shown in Figure 3A. At high ionic strengths (>10^−4^ M), conductance measurements displayed a linear dependence on the ionic strength. In these conditions, the Debye length was significantly smaller than the size of nanochannels. Accordingly, the results were consistent with the bulk electrolyte conductance (red dashed line in Figure 3A). In contrast, at low ionic strengths (≤10^−4^ M), we observed a plateau in conductance (in the log-log scale). This occurred when the Debye length approached the nanochannel dimension, and the excess of counter-ions balanced the surface charge, reaching channel electroneutrality [68]. Here, as it directly related to the conductance, the surface charge could be calculated by fitting the results to the equation [69]:(1)$$\frac{I}{V}=2F\mu \sqrt{{\left(\frac{\Sigma }{2}\right)}^{2}+c_{0}^{2}}\frac{wh}{l}$$

In Equation (1), F, μ, and Σ are the Faraday’s constant, ionic mobility, and the volume charge density, respectively. Further, $c_{0}$ is the solution molarity, and w, h, and l are the nanochannels’ width, height, and length, respectively. Using the relation $zF\Sigma =-2\sigma_{s}/h,$ we obtained a surface charge value of $\sigma_{s}=1.81$ µC/m^2^, which was consistent with the previously reported data for SiC surfaces [70]. Our SiC coating exhibited a surface charge orders of magnitude smaller than SiO_2_ (1–100 mC/m^2^) [71], which correlated with better performance in electrostatic gating control. In fact, chemically reactive surfaces act as charge buffers. An externally applied electric field is quickly compensated by protonation or deprotonation of reactive groups on the surface, limiting charge rearrangement in the electrical double layer (EDL) [72]. Thus, to minimize surface charge, materials are often artificially treated [28].

### 3.4. Electrochemical Characterization: Electrochemical Impedance Spectroscopy

To investigate dielectric/liquid interface properties with the application of an external voltage, we performed electrochemical impedance spectroscopy (EIS) measurements. Specifically, we compared the resistance to charge transfer and the double layer capacitance at different gate voltages. A comparative assessment was conducted using SiC- and SiO_2_-coated chips. Figure 3B shows a schematics of the electrical double layer (EDL), which described the ionic distribution that occurres at the solid–liquid interface of a charged surface to maintain local electroneutrality. The EDL is usually described by that Grahame model, which identifies three main layers consisting of 1) non-hydrated ions adsorbed to the surface, ii) hydrated immobile ions, iii) free moving hydrated ions [73]. The first and second layers of immobile ions are often referred to as the Stern layer. The EDL region is modeled by a series of capacitors, referred to as double-layer capacitance (C_EDL_), where the Stern layer (~0.2 nm) [74] corresponds to the most significant contribution. As current can flow across the interface upon application of a DC potential, a resistive path is considered in parallel to the capacitance. This is usually referred to as a charge-transfer resistance (R_ct_). R_ct_ can vary substantially depending on the material ability to exchange electrons with the electrolyte solution. Upon application of an external DC potential, if electrons cannot be easily exchanged, an overpotential builds up at the interface. In non-polarizable materials, such as Ag/AgCl, small R_ct_ permits high currents. In contrast, polarizable materials present high R_ct_, and the current exchange is limited.

By fitting our EIS measurements to the model described above (Figure 3B), we calculated R_ct_ and C_EDL_ for both a SiO_2_- and a SiC-coated membrane at different gate voltages (V_G_) applied (Figure 3C,D). Depending on the applied potential, SiC showed an R_ct_ 1.5 to 8 times bigger than SiO_2_ (Figure 3C). Interestingly, both materials showed a clear dependence of R_ct_ with the applied voltage. SiO_2_ exhibited a monotonic increase of R_ct_ with the applied voltage, where more positive voltages resulted in higher R_ct_. In contrast, SiC showed a decrease in R_ct_ proportional to the absolute value of the applied voltage. We attributed these phenomena to the difference in surface charge between SiO_2_ and SiC. In fact, a higher number of available SiO^−^ sites on the SiO_2_ surface allowed for increased electron exchange.

C_EDL_ did not exhibit a correlation with the applied gate voltage for either material (Figure 3D). Moreover, we unexpectedly found six times higher C_EDL_ for SiC with respect to SiO_2_. As C_EDL_ mainly depends on the surface area and EDL thickness, our results could be explained in the context of the material porosity [75]. By presenting a larger surface area, pores displayed increased capacity. Overall, the low surface charge exposed by SiC and the high resistance to charge transfer qualified SiC as a polarizable interface suitable for electrostatic gating.

### 3.5. Mechanism of Analyte Flow Control through Electrostatic Gating

Nanofluidic systems present high surface to volume ratios. In light of this, charged species diffusing in nanoconfinement exhibit unique behaviors [76,77]. Electrostatic, steric, and hydrodynamic interactions with the nanochannel walls influence local molecular concentration and effective diffusivity. Depending on solution properties, such as ionic strength, pH, and surface charge density, the EDL can extend from a fraction to hundreds of nm in the fluid. Both SiC and SiO_2_ surface expose native silanol groups, resulting in a net negative surface charge at pH 7.4 [78]. In proximity to the surface, charged species redistribute to reach electroneutrality [73]. While counter-ions concentration increases, co-ions are depleted following distribution with a characteristic dimension equal to the Debye length.

Once solution properties are defined, the surface charge is the only parameter that has a significant effect on the distribution of charges in the fluid. Thus, nanochannel charge-selectivity can be altered by controlling the channel surface charge. An applied difference in potential between a buried gate electrode and an electrode in solution creates an overpotential at the surface. We employed this strategy to modulate the diffusive transport of analytes through our nanofluidic membrane. With no applied voltage, molecules diffused through the channel unperturbed. By applying a negative gate potential, the transmembrane transport of co-ions was substantially reduced.

### 3.6. In Vitro Release Modulation of Methotrexate

To investigate the effectiveness of electrostatic gating on controlling trans-membrane transport of a small charged analyte, we performed an in vitro diffusion study using methotrexate. Methotrexate has a molecular weight of 454 Da and is a good representative of small molecules (<900 Da) therapeutics, which accounts for the majority of pharmaceuticals [79]. Clinically, methotrexate is used as a chemotherapeutic agent for the treatment of various cancers, as well as in the management of rheumatoid arthritis [33].

Figure 4A shows the normalized release rates for four consecutive cycles alternating between passive and active phases. During the passive phases, negatively charged molecules (−2q for methotrexate) diffused trough the nanochannels freely, largely unaffected by the low native charge of the SiC surfaces. When a negative voltage was applied (−3 V), an increase in negative surface charge repelled methotrexate molecules, reducing their release. The four alternation cycles between passive and active phases demonstrated that electrostatic gating allowed for repeatability of release modulation.

We observed a statistically significant (** *p* ≤ 0.01) difference in release rate between active and passive phases, whereby the applied potential −3 V yielded a decrease in the release rate of ~35%. During the passive phase, an average release rate of 10 µg/day was obtained, which was consistent with daily doses used to treat rheumatoid arthritis in pre-clinical testing [80]. Other small molecule therapeutics, including glucocorticoids [81], hormone therapeutics [82], and antivirals [83], present effective daily doses in the order of micrograms. This indicates that the current membrane architecture could, in principle, be adopted for various therapeutic applications. However, further testing with different pharmaceutical agents is warranted.

### 3.7. In Vitro Controlled Release of Quantum Dots

To assess the ability of our membrane to modulate the release rate of larger molecules, we performed an in vitro release study with quantum dots. Quantum dots possess broad applicability in bioengineering, including imaging [84], theranostics [85], cell labeling for in vivo tracking [86], tissue staining [87]. They have also been investigated as biomarkers for cancer detection and for targeted drug delivery [35]. Figure 5A shows the normalized release rate of each phase, where passive (0 V) and active phases (−1.5 V) were alternated over three cycles. 

The application of the negative gate potential drastically reduced the release of quantum dots from the membrane. Subsequent cycles demonstrated consistent and reproducible release rate reduction, suggesting that the membrane and the gating performance were consistent over time. A statistically significant difference (**** *p* ≤ 0.0001) in the release between active and passive phases (84%) was observed (Figure 5B). When compared to methotrexate, quantum dots clearly showed a more effective electrostatic modulation, which could be attributed to higher particle charge and lower ionic strength of the solution. Specifically, the high exposed charge is due to the carboxylic functionalization, where several groups result in a negative net charge that ranges from −5 to −15 depending on pH and ionic strength [41]. Moreover, the low ionic strength solution (0.01 × PBS) has a Debye length 10 times greater than the 1 × PBS. These two properties contribute to enhance the electrostatic interactions between the wall and the solute. Thus, the application of the gate potential resulted in increased efficacy of release modulation.

### 3.8. Considerations on Electrostatic Gating Performance

To achieve efficient devices for tunable molecular diffusion via electrostatic gating, various parameters need to be optimized. Of utmost importance is the choice of dielectric material to insulate the buried gate electrode. In this study, we investigated SiC as it conciliates the need for low leakage currents, with a dielectric constant similar to SiO_2_ (4.4–4.9) [88], and offers chemical inertness in aqueous solutions [48]. Moreover, SiC offers a low native charge; therefore, it minimizes unwanted non-linearities connected to the buffer capacity of strongly charged surfaces [72].

Further, efficient electrostatic flow modulation is strictly connected to the nanochannel size to the Debye length ratio (h/λ). Our membrane was designed for medical applications, where the ionic strength and pH were bound to physiological values. Our future investigations focus on manufacturing membranes with smaller nanochannels to be able, in principle, to completely stop analyte diffusion. Finally, as flow control trough electrostatic gating is mainly based on coulombic interactions, analytes that expose high surface charges are more suitable for gate modulation. Therefore, drug encapsulation with highly charged polymers can significantly improve administration control of small analytes.

## 4. Conclusions

In this work, we investigated a SiC-coated nanofluidic membrane capable of the reproducible control of analyte transport via electrostatic gating. The application of a low-intensity electrical potential to the gate electrode allowed us to alter nanochannel surface charge, leading to tunable membrane charge-selectivity, and control over the release of methotrexate and quantum dots. Electrochemical characterization showed that SiC dielectric coating exhibited low leakage current and reduced intrinsic charge as compared to SiO_2_. Moreover, SiC offered chemical bioinertness, which rendered it an ideal candidate for use in biomedical devices for therapeutic delivery based on electrostatic-gating. In this context, our membranes could be employed as actuators for remotely controlled drug delivery systems. The low voltage needed to modulate the release rate could be provided via small scale and low-power circuitry. This investigation might pave the way for the development of the next generation of drug delivery systems, enabling pre-programmed or remotely managed pharmaceutic administration. Further, our gated nanofluidic membrane might find applicability in molecular sieving and lab on a chip diagnostic.

## 5. Patents

Grattoni, A.; Liu, X.; Ferrari, M. Gated Nanofluidic Valve For Active And Passive Electrosteric Control Of Molecular Transport, And Methods Of Fabrication, U.S. Provisional Pat. Ser. No. 62/961,437, filed Jan 15. (2020).

## Figures and Tables

**Figure 1 pharmaceutics-12-00679-f001:**
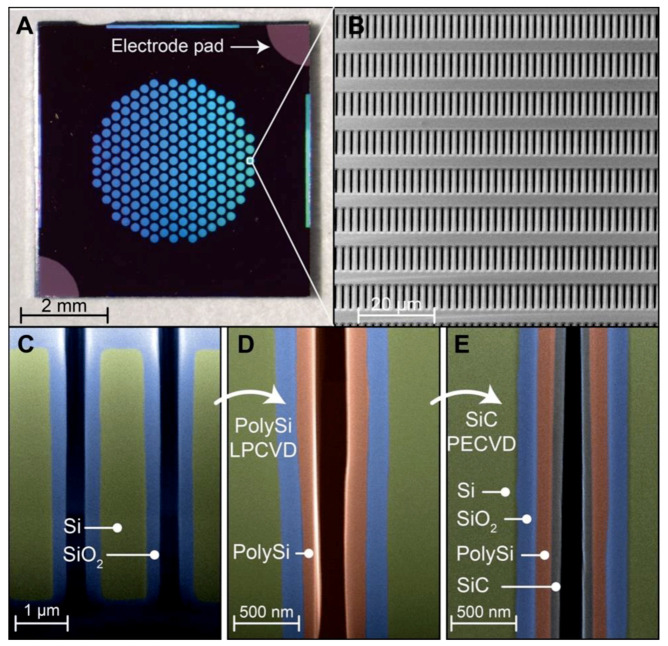
Nanochannel membrane structure. (**A**) Optical image of a silicon nanofluidic membrane, presenting electrode pads with exposed conductive polysilicon. (**B**) SEM micrograph, showing the array of nanochannel inlets. (**C**,**D**,**E**) Vertical cross-section image (SEM) obtained along the length of nanochannel, showing the membrane fabrication at different stages. Micrographs were color-enhanced for clarity of visualization. (**C**) Thermally grown SiO_2_ layer (~175 nm, blue); (**D**) Low-pressure chemical vapor deposition (LPCVD)-deposited poly-Si layer (~121 nm, red); (**E**) Plasma-enhanced-CVD deposited SiC coating (~64 nm, gray). Images **C**, **D**, and **E** do not picture the same membrane location.

**Figure 2 pharmaceutics-12-00679-f002:**
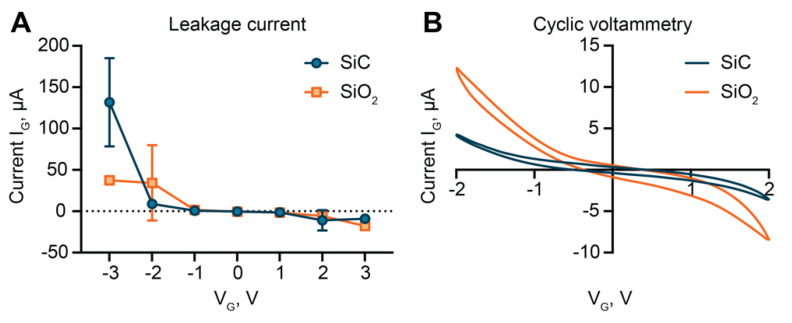
Leakage current and cyclic voltammetry. (**A**) Comparison of gate leakage current for SiO_2_ and silicon carbide (SiC) dielectric. (**B**) Cyclic voltammetry comparison between SiO_2_ and SiC.

**Figure 3 pharmaceutics-12-00679-f003:**
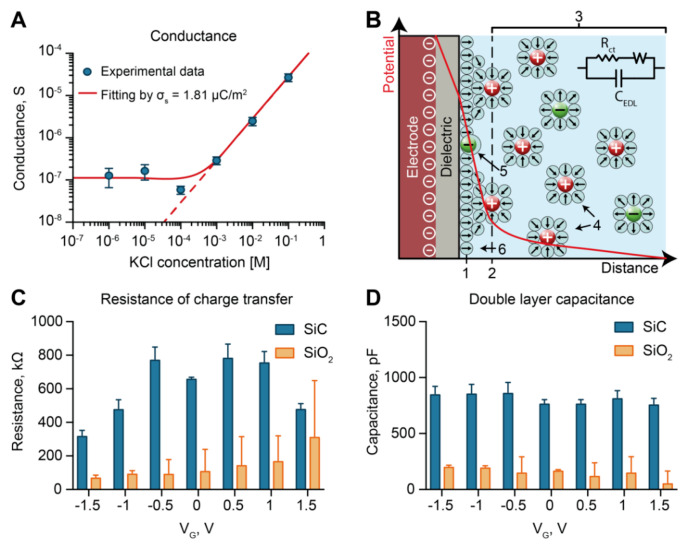
Electrochemical measurements. (**A**) Measured transmembrane ionic conductance. (**B**) Schematic of the electric double layer and relative model. (1) Inner Helmholtz plane; (2) Outer Helmholtz plane; (3) Diffuse layer; (4) Solvated ion; (5) Specifically adsorbed ion; (6) Molecules of the electrolyte solvent. (**C**) Fitted resistance of charge transfer (R_ct_) of SiO_2_-coated membranes versus SiC-coated membranes. (**D**) Fitted double-layer capacitance (C_dl_) of SiO_2_-coated membranes versus SiC-coated membranes.

**Figure 4 pharmaceutics-12-00679-f004:**
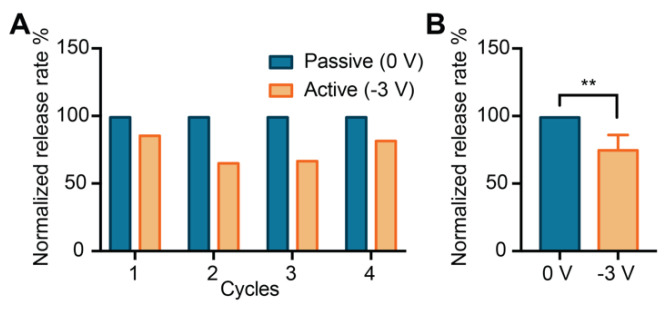
Electrostatically controlled release of methotrexate. (**A**) The normalized release rate of methotrexate for four cycles between free diffusion (Passive) and gated diffusion (Active). (**B**) Release rates grouped by phase typology (** *p* ≤ 0.01).

**Figure 5 pharmaceutics-12-00679-f005:**
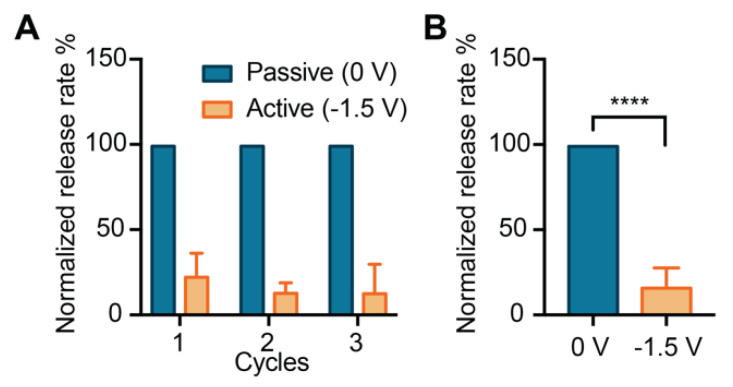
Electrostatically controlled release of quantum dots. (**A**) The normalized release rate of quantum dots for three cycles between free diffusion (Passive) and gated diffusion (Active). (**B**) Release rates grouped by phase typology (**** *p* ≤ 0.0001).

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
