# Peer review of "Silicon Nanofluidic Membrane for Electrostatic Control of Drugs and Analytes Elution"

_pharmaceutics, 2020, doi:10.3390/pharmaceutics12070679_

Round 1

Reviewer 1 Report

This article presents a nanofluidic membrane for controlled actuable molecular transport for drug delivery applications. The paper is well articulated and presented. However, there are a few minor issues that require fixing before acceptance of this work.

  1. Methods section could be improved for reproducibility.
  2. The authors claim that the materials they have used are more biocompatible compared to the ones that are generally used for making electrical devices. However, now data on cell adhesion or viability is provided after each step of fabrication.
  3. What is the biofouling potential of the prepared device, it should be discussed in the manuscript.
  4. The discussion relating to the better control over transport of quantum dots compared to methotroxate is very weak and should be expanded.
  5. Despite a large number of references in the list, I believe some are redundent and other key references in this field are missing. I recommend addition of these references: https://onlinelibrary.wiley.com/doi/abs/10.1002/adma.201102090  https://pubs.acs.org/doi/abs/10.1021/ac201636n https://onlinelibrary.wiley.com/doi/abs/10.1002/adma.201500473 https://pubs.acs.org/doi/abs/10.1021/am4013984 https://www.sciencedirect.com/science/article/abs/pii/S0008622315004972

Reviewer 2 Report

This is a very complex work with great effort invested by the authors in order to accomplish their results.

Some important though minor additions are required to improve the quality of this paper:

  1. The procedure to obtain the  nanofluidic membrane system israther complex. Please corroborate the readers' comprehension with a schematic showing the device manufacturing steps as described in materials and methods.
  2. The schematic as required in (1) can be also accompanied by SEM/AFM/TEM or other evidence of the system components in the same or another figure.
  3. The measurements algorithm/procedure can also be given in another schematic.
  4. Quantum Dots where obtained from Sigma as mentioned by the authors. A structural photo (TEM etc) would help since so other companies deliver these. What are their mean diameters?
  5. Finally, can the authors justify the nanofluidic characterization of their device since, in order to be in the nano - scale a material or device has to be below the 100nm border. The channels of the device are not, nor the membrane size. So, maybe a microchannels and microfluidics terminology suits better this device ? This affects also the paper title.
